# Effective Young's Modulus of Complex Three Dimensional Multilayered Ti/Au Micro-Cantilevers Fabricated by Electrodeposition and the Temperature Dependency

Hitomi Watanabe [1], Tso-Fu Mark Chang [1,*], Michael Schneider [2], Ulrich Schmid [2], Chun-Yi Chen [1], Shinichi Iida [3], Daisuke Yamane [4], Hiroyuki Ito [1], Katsuyuki Machida [1], Kazuya Masu [1] and Masato Sone [1]

1    Institute of Innovative Research, Tokyo Institute of Technology, Kanagawa 226-8503, Japan; watanabe.h.av@m.titech.ac.jp (H.W.); chen.c.ac@m.titech.ac.jp (C.-Y.C.); ito@pi.titech.ac.jp (H.I.); machida.k.ad@m.titech.ac.jp (K.M.); masu.k.aa@m.titech.ac.jp (K.M.); sone.m.aa@m.titech.ac.jp (M.S.)
2    Institute of Sensor and Actuator Systems, Vienna University of Technology, Gußhausstraße 27-29, 1040 Wien, Austria; michael.schneider@tuwien.ac.at (M.S.); ulrich.e366.schmid@tuwien.ac.at (U.S.)
3    NTT Advanced Technology Corporation, Atsugi 243-0124, Kanagawa, Japan; shinichi.iida@ntt-at.co.jp
4    Department of Mechanical Engineering, Ritsumeikan University, 1-1-1 Noji-Higashi, Kusatsu 525-8577, Shiga, Japan; dyamane@fc.ritsumei.ac.jp
*    Correspondence: chang.m.aa@m.titech.ac.jp; Tel.: +81-45-924-5631

**Abstract:** Ti/Au multi-layered micro-cantilevers with complex three-dimensional structures used as micro-components in micro-electromechanical systems (MEMS) sensors were prepared by lithography and electrodeposition, and the effective Young's modulus was evaluated by the resonance frequency method and finite element method simulation. Effects of the constraint condition at the fixed-end of the micro-cantilever and the temperature dependency of the effective Young's modulus were studied. Three types of the constraint at the fixed-end were prepared, which were normal type (constraining only bottom surface of the fixed-end), block type (constraining both top and bottom surfaces), and bridge type (top surfaces covering with a bridge-like structure). The temperature dependency test was conducted in a temperature range from 150 to 300 °C in a vacuum chamber. An increase in the effective Young's modulus was observed as the constraint condition became more rigid, and the effective Young's modulus merely changed as the temperature varied from room temperature to 300 °C.

**Keywords:** effective Young's modulus; multi-layered structure; Ti/Au; electrodeposited gold; micro-cantilever; resonance frequency method; FEM

## 1. Introduction

Size reduction and performance enhancement of micro-electromechanical systems (MEMS) devices are always challenging and important research interests, especially for MEMS devices, such as MEMS accelerometers commonly equipped in smartphones. Recently, the application of MEMS accelerometers for biomedical applications has been proposed [1–3]. Detection of body tremors [3] and muscular sounds [4] has been reported using a highly sensitivity MEMS accelerometer. Early detection of neurological intractable diseases such as Parkinson's diseases would be possible by analyzing information obtained from the body tremors and muscular sounds.

Size reduction of MEMS devices having the sensitivity greatly affected by the noise level, such as Brownian noise, is difficult since the noise level is dependent on the overall mass of the key components, and a number of mass or volume is needed to maintain a low noise level or high sensitivity. Precious metal materials have excellent properties, such as high chemical stability, electrical conductivity, and biocompatibility, and are advantageous toward electronic devices [5]. Recently, the application of gold-based materials to MEMS devices has received a great amount of attention, and MEMS accelerometers employing

gold components prepared by electrodeposition are reported to have a small size whiling retaining high sensitivity [2,3], which is mainly contributed by gold's high mass density ($19.3 \times 10^3$ kg/m$^3$ at 298 K [6]). The acceleration sensing unit used for accelerometers is G (1 G $\approx$ 9.8 m/s$^2$). The sensing ability of a gold-based MEMS accelerometer is reported to reach μG level and the Brownian noise is merely 22 nG/$\sqrt{Hz}$ [3].

On the other hand, gold is known to be a relatively soft metallic material among materials commonly utilized in electronics, and electrodeposition is a powerful method to strengthening the gold by grain refinement [7,8] alloying [9–11] and introduction of nanotwins [12]. Alternately and repeatedly deposition of gold following by a material with higher mechanical strength to realize a multilayered structure is also an effective strategy [13–15]. Cantilever-like structures are commonly used in movable components in MEMS devices. From the Euler-Bernoulli beam theory [16], structural stability of a cantilever can be improved by using materials having high Young's modulus. The Young's modulus of gold is 78.5 GPa [8], which is much lower than that of copper (128 GPa) and silicon (165 GPa) [9]. Copper and silicon are materials commonly used in electronic components. Thus, a multilayered structure composed of titanium, having a high Young's modulus at 120 GPa [8] and electrodeposited gold is proposed for movable components in MEMS devices. The titanium layer also serves to improve adhesion of the gold layer on the silicon substrate and the barrier layer to prevent the diffusion of gold into the substrate.

In mechanical property evaluation of materials toward miniaturized electronics, the mechanical property is reported to be affected by the size of the specimen used in the evaluation, which is known as the sample size effect [17–19]. However, the Young's modulus is an intrinsic property of materials, which should be constant as the specimen size changes, but the Young's modulus of small-sized specimens is reported to be different as the size changes, and the Young's modulus of a small-sized specimen with a specific geometry is called the effective Young's modulus [20]. The effectively Young's modulus of micro-cantilevers could be determined by a non-destructive method by determining the resonance frequency of the micro-cantilever with a laser doppler vibrometer [20,21] then the effective Young's modulus can be calculated from the resonance frequency.

Gold based MEMS has a complex three dimensional (3D) multilayered Ti/Au structures [2] and the annealing process is often applied in the fabrication of electronic devices, such as 310 °C treatment in fabrication of MEMS accelerometers [13], and the thermal energy could greatly affect the properties of the material and performance of the device [22,23]. Hence, temperature dependency of the effective Young's modulus of micro-cantilevers composed of electrodeposited gold and sputtered titanium is evaluated in this study. In addition, the 3D multilayered Ti/Au structures are related to the constraint condition at the fixed-end of micro-cantilevers, which could affect the structure stability of micro-cantilevers. Therefore, effects of the constraint condition on the effective Young's modulus are also studied.

## 2. Experimental

### 2.1. Structure Design and Fabrication Process of the Ti/Au Multilayered Micro-Cantilever

Two types of the Ti/Au multilayered structure were evaluated in this study. A schematic view of the Ti/Au multilayered cantilever is shown in Figure 1. One type was a Ti/Au single layered, and one was double layered. The substrate was a SiO$_2$ coated silicon substrate. The thickness of the SiO$_2$ layer was about 0.5 μm, and the layer was formed by sputtering. The titanium layer was deposited on top of the substrate by sputtering with a designed thickness of 100 nm, and a thin gold layer was also sputtered to be used as the seed layer in the later gold electrodeposition process. The designed length (*l*) was 200, 600, and 1000 μm, and the width (*w*) was 15 μm. Thickness of one electrodeposited gold layer was designed to be 10 μm. The micro-cantilevers were then manufactured by repeated processes of lithography and electrodeposition [24]. The gold electrolyte was a sulfite-based electrolyte containing 50 g/L of Na$_2$SO$_3$, 50 of g/L (NH$_4$)$_2$SO$_3$, 10 g/L of Au,

and 5% sodium gluconate with pH of 8.0. A 310 °C annealing process was conducted in the fabrication process.

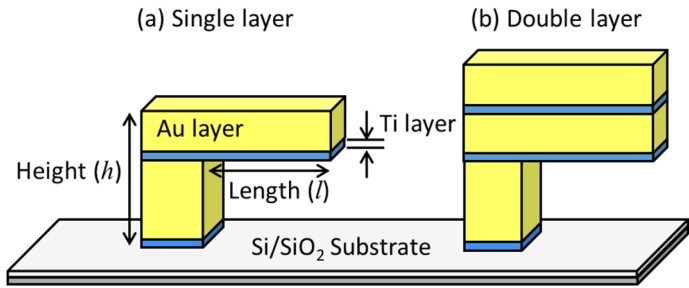

**Figure 1.** Schematics of the Ti/Au (**a**) single and (**b**) double layered micro-cantilevers.

Three types of the fixed-end condition of the micro-cantilever were studied as shown in Figure 2. These types of complex 3D multilayered Ti/Au structure were used as micro-component structures in gold-based MEMS sensors [2,3]. The normal type was with a constraint on the bottom surface of the fixed-end only. The block (BL) type was with constraints on both the top and the bottom surface of the fixed-end. Bridge (BR) type was with a bridge-like structure on top of the fixed-end.

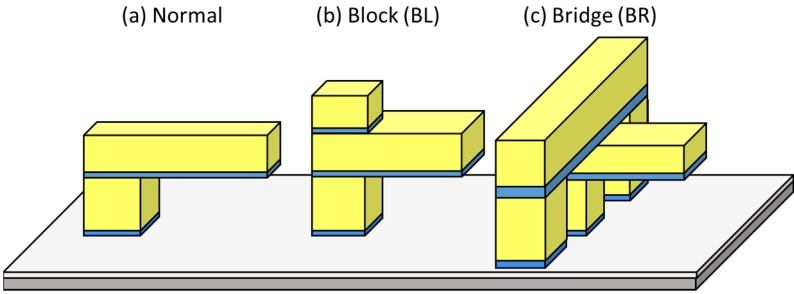

**Figure 2.** Schematics of the (**a**) normal, (**b**) block and (**c**) bridge type micro-cantilevers.

### 2.2. Temperature Dependency Test

The as-fabricated micro-cantilevers were annealed in a vacuum chamber (Heraeus Vacutherm Compact VT 6025, Thermo Scientific, Great Britain, UK) at various temperatures. The annealing temperatures were 150, 200, 250, 300 °C. During the annealing process, the micro-cantilevers were placed in the vacuum chamber at the specific annealing temperature for 30 min. Then, the temperature was lowered back to the room temperature, and the sample was left in the vacuum chamber for at least one and a half hour.

### 2.3. Effective Young's Modulus by Resonance Frequency Method

After the annealing process, the effective Young's modulus of the micro-cantilevers was evaluated by determining the resonance frequency using a laser doppler vibrometer (LDV, MSA-400, Polytec, Waldbronn, Germany), as shown in Figure 3. Exact dimensions of the micro-cantilevers were confirmed using a scanning electron microscope (SEM, SU4300SE Hitachi Co., Ltd., Tokyo, Japan) and a 3D optical microscope (OM, VHX-5000, Keyence, Osaka, Japan) equipped with a 3D display and a 3D measurement function for calculation of the effective Young's modulus.

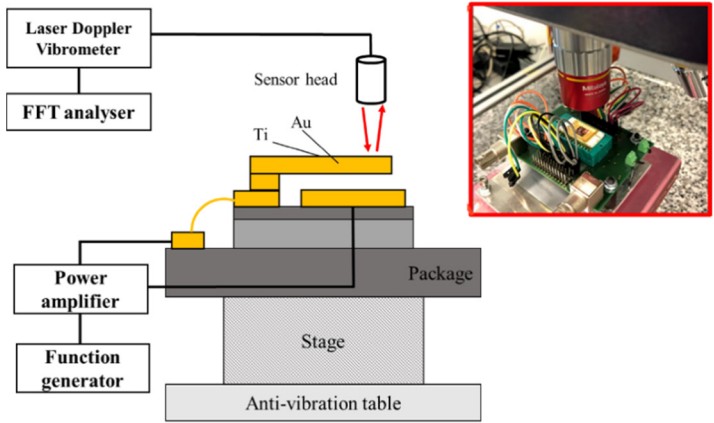

**Figure 3.** Resonance frequency measurement system.

*2.4. FEM Simulation of the Resonance Frequency*

Finite element method (FEM) simulations were performed using simulation software (COMSOL Multiphysics) to analyze the resonance frequency of the Ti/Au multilayered micro-cantilevers at different temperatures and constraint conditions at the fixed-end. The micro-cantilevers analyzed consisted of two sections, namely the fixed-end and the beam body. The length and width of the beam body were set at 200 μm and 15 μm, respectively, while the width of the fixed-end was set at 30 μm. For the normal type, only the lower surface of the fixed-end was restrained. For the block type, both the upper and lower surface of the fixed-end were restrained. For the bridge type, the upper surface of the fixed-end was restrained using a bridge-like structure. The constants of linear elasticity, such as Young's modulus, thermal expansion coefficient, Poisson's ratio, and density, were applied to the simulation. These constants were provided by the database incorporated in COMSOL Multiphysics [25]. The Young's modulus of titanium is 115.7 GPa and the Young's modulus of gold is 70 GPa, which are intrinsic values of the bulk material. The FEM analysis was carried out based on the conditions.

**3. Results and Discussion**

SEM images of the micro-cantilevers with the block and bridge type constraint at the fixed-end are shown in Figure 4. Multiple Ti/Au layered structure piled up on the top surface of the fixed-end of the micro-cantilever could be observed at fixed-end of the block type micro-cantilevers as shown in Figure 4a. For the bridge type, a complex bridge-like structure on the top surface with two additional columns supporting the body of the bridge could be observed, as shown in Figure 4b.

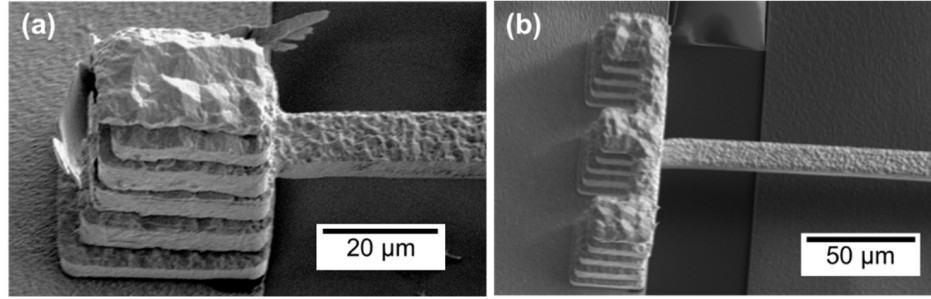

**Figure 4.** SEM images of the (**a**) block and (**b**) bridge type Ti/Au micro-cantilevers.

A relationship between the effective Young's modulus ($E_{\mathrm{eff}}$) and the resonance frequency ($f_c$) could be expressed by the following equation [20]:

$$f_{\mathrm{c}} = 0.16154 \frac{h}{L^2} \sqrt{\frac{E_{\mathrm{eff}}}{\rho}},$$ (1)

where $h$ and $L$ and are the thickness and length of the fabricated Ti/Au micro-cantilevers, respectively. $\rho$ is the density of gold (19.3 g/cm$^3$). The thickness of the titanium layer was much smaller than that of the gold layer. Therefore, the mass density and thickness of the titanium layer were assumed to have negligible influence on the resonance frequency. For calculation of the effective Young's modulus, the equation could be rearranged as follows:

$$E_{\mathrm{eff}} = \rho \left( \frac{f_{\mathrm{c}} L^2}{0.16154 h} \right)^2,$$ (2)

The effective Young's moduli of the as-fabricated Ti/Au micro-cantilevers with different number of the Ti/Au layered structure and the fixed-end condition are summarized in Table 1. A decrease in the effective Young's modulus following an increased in the number of the Ti/Au layer was observed. For the normal type Ti/Au micro-cantilever, the effective Young's modulus decreased from 62.4 to 45.0 GPa when the Ti/Au layered structure increased from single layer to triple layer. This finding revealed the size effect observed in the Young's modulus of specimens in small size. On the other hand, as shown in Equation (2), the thickness is already considered in calculation of the effective Young's modulus, hence thickness dependency of the effective Young's modulus is not expected as the thickness increases. Therefore, this change in the effective Young's modulus observed in this study is suggested to be a result of the multilayered structure. That is to say, the titanium layer(s) sandwiched between the gold layers could have an influence on the resonance frequency.

**Table 1.** Effective Young's modulus of the as-fabricated Ti/Au micro-cantilevers by resonance frequency method.

| Fixed-End | Single Layer | Double Layer | Triple Layer |
|---|---|---|---|
| Normal | 62.4 | 52.4 | 45.0 |
| Block | 62.7 | 57.4 | 48.0 |
| Bridge | 63.3 | 58.0 | 49.5 |

[GPa].

Regarding the effects of the fixed-end condition, a light increase in the effective Young's modulus was obtained as the constraint condition at the fixed-end became more rigid. The rigidness was suggested to be normal < block < bridge. For the Ti/Au single layered micro-cantilevers, the effective Young's modulus increased from 62.4 to 62.7 and 63.3 GPa as the fixed-end changed from the normal to the block and bridge type, respectively. Considering the report by Teranishi et al. [15], the structural stability of the cantilever increases as the fixed-end condition becomes more rigid. Slight deformations on the top surface at the fixed are observed for the normal type fixed-end when applying forces to the micro-cantilever. As the micro-cantilever vibrates during the resonance frequency measurement, the deformation at the fixed-end end would affect the overall resonance frequency of the micro-cantilever. Effects of the fixed-end condition on the effective Young's modulus were also observed in the Ti/Au single layered micro-cantilevers from the FEM simulation, as shown in Table 2, where the effective Young's modulus increased from 68.1 to 69.4 GPa as the fixed-end changed from the normal to the block and bridge type.

**Table 2.** Effective Young's modulus of the as-fabricated Ti/Au micro-cantilevers by FEM simulations.

| Fixed-End | Single Layer |
|-----------|--------------|
| Normal    | 68.1 GPa     |
| Block     | 69.4 GPa     |
| Bridge    | 69.4 GPa     |

Table 3 shows the effective Young's modulus of Ti/Au single layered micro-cantilevers annealed at 150, 200, 250, and 300 °C for 30 min. At first, the effective Young's moduli of the micro-cantilevers with a more rigid fixed-end were all higher at all temperatures, which corresponds well with the results obtained from the as-fabricated Ti/Au micro-cantilevers. A less than 1% increase in the effective Young's modulus was observed as the temperature increased from 150 to 300 °C in the normal, block, and bridge type micro-cantilevers. The increases were small, but all three types of the micro-cantilever showed the same trend. The increase is suggested to be caused by the formation of Ti-Au intermetallic compounds at the Ti/Au interface [26,27]. At an elevated temperature, diffusions of titanium and gold elements would be enhanced to promote the formation of the intermetallic compounds, and eventually cause an increase in the Young's modulus.

**Table 3.** Effective Young's modulus of the annealed Ti/Au single layered micro-cantilevers by resonance frequency method.

| Fixed-End | 150 °C | 200 °C | 250 °C | 300 °C |
|-----------|--------|--------|--------|--------|
| Normal    | 62.3   | 62.3   | 62.3   | 62.5   |
| Block     | 62.9   | 63.0   | 63.1   | 63.3   |
| Bridge    | 65.1   | 65.1   | 65.3   | 65.5   |

[GPa].

## 4. Conclusions

In this study, the electrodeposited Ti/Au multilayered micro-cantilevers with different fixed-end conditions were fabricated as examples of the complex 3D multilayered Ti/Au structure used as component structures in gold-based MEMS sensors. The effective Young's modulus of the Ti/Au multilayered micro-cantilevers with different fixed-end conditions and the temperature dependency were reported. The effective Young's modulus obtained from the resonance frequency method showed a decreasing trend of the number of the Ti/Au layers increased and an increasing trend as the fixed-end varied from the normal to block and bridge types. A similar increasing trend in the effective Young's modulus as the fixed-end became more rigid was obtained from the FEM simulations. In addition, an increase in the effective Young's modulus was observed as the annealing temperature increased from 150 to 300 °C. The fixed-end conditions of movable components and the annealing temperatures evaluated in this study are commonly used in the fabrication of miniaturized electronics, and the findings reported here could contribute to the design of MEMS components in future.

**Author Contributions:** Conceptualization, H.W. and T.-F.M.C.; methodology, H.W.; software, H.W.; validation, C.-Y.C. and S.I.; formal analysis, H.W. and T.-F.M.C.; investigation, H.W. and T.-F.M.C.; resources, T.-F.M.C., M.S. (Michael Schneider), and U.S.; data curation, H.W.; writing—original draft preparation, H.W.; writing—review and editing, T.-F.M.C.; visualization, H.W. and T.-F.M.C.; supervision, T.-F.M.C. and M.S. (Masato Sone); project administration, D.Y., H.I., and K.M. (Katsuyuki Machida); funding acquisition, K.M. (Kazuya Masu) and M.S. (Masato Sone). All authors have read and agreed to the published version of the manuscript.

**Funding:** This study is based on results obtained from a project commissioned by the New Energy and Industrial Technology Development Organization (NEDO), and is supported by JST CREST Grant Number JPMJCR1433.

**Institutional Review Board Statement:** N/A.

**Informed Consent Statement:** N/A.

**Data Availability Statement:** The data presented in this study are available on request from the corresponding author.

**Conflicts of Interest:** On behalf of all of the co-authors, the corresponding author states that there is no conflict of interest.

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
