# Peer review of "Effective Young’s Modulus of Complex Three Dimensional Multilayered Ti/Au Micro-Cantilevers Fabricated by Electrodeposition and the Temperature Dependency"

_2673-3293, doi:10.3390/electrochem2020015_

Round 1
Reviewer 1 Report
In this paper, Ti/Au multi-layered micro-cantilevers with complex three-dimensional structures used as 17 micro-components in MEMS sensors were successfully prepared by lithography and electrodeposition.
The innovativeness of the preparation of this article and the combination of the two technologies impressed me. My suggestion is to accept and publish it after making the following minor revision.
Comments:
- In the 50th sentence, the root sign is not fully displayed.
- In the 94th sentence, “The substrate was a SiO2 coated silicon substrate”, please the coated process should be introduced in detail.
- In the 100th sentence, “A 310 °C annealing process was conducted in the fabrication process”. Is this process performed under vacuum or under protective atmosphere?
- The electrodeposition solution used in this study should be described in detail.
- The pH value of electrodeposition solution should be described in detail.
- In the preparation process of this article, the design thickness of Au is 10um, but the thickness of the actual sample is less than 10um. Please explain its reason.
- In equation 2, the comma should be deleted.
- Since micro-electrodeposition technology is employed in the paper, the following related papers should be cited in the introduction to electrochemical deposition (9-12).
- Electrochemical Behavior and Electrodeposition of Sn Coating from Choline Chloride-Urea Deep Eutectic Solvents.
- Direct-Write Printing Copper–Nickel (Cu/Ni) Alloy with Controlled Composition from a Single Electrolyte Using Co-Electrodeposition.
- A Hybrid process for printing pure and High conductivity nanocrystalline copper and nickel on flexible polymeric Substrates.
- Additive printing of pure nanocrystalline nickel thin films using room environment electroplating

Author Response
Comment
In this paper, Ti/Au multi-layered micro-cantilevers with complex three-dimensional structures used as 17 micro-components in MEMS sensors were successfully prepared by lithography and electrodeposition.
The innovativeness of the preparation of this article and the combination of the two technologies impressed me. My suggestion is to accept and publish it after making the following minor revision.
Response
Thank you for the positive comments. We have revised the manuscript based on the comments.
Comment #1
In the 50th sentence, the root sign is not fully displayed.
Response #1
The root symbol has been revised.
Revision made
Line 50
………and the Brownian noise is merely [3].
Comment #2
In the 94th sentence, “The substrate was a SiO2 coated silicon substrate”, please the coated process should be introduced in detail.
Response #2
More details of the SiO2 layer has been added.
Revision made
Line 94
Thickness of the SiO2 layer was about 0.5 μm, and the layer was formed by sputtering.
Comment #3
In the 100th sentence, “A 310 °C annealing process was conducted in the fabrication process”. Is this process performed under vacuum or under protective atmosphere?
Response #3
The annealing was conducted under atmospheric condition.
Comment #4
The electrodeposition solution used in this study should be described in detail.
Response #4
Composition of the electrolyte has been added to the experimental section.
Revision made
Line 100-102
The gold electrolyte was a sulfite-based electrolyte containing 50 g/L of Na2SO3, 50 of g/L (NH4)2SO3, 10 g/L of Au, and 5 % sodium gluconate with pH of 8.0.
Comment #5
The pH value of electrodeposition solution should be described in detail.
Response #5
pH of the electrolyte was 8.0. The information has been added to the experimental section.
Comment #6
In the preparation process of this article, the design thickness of Au is 10um, but the thickness of the actual sample is less than 10um. Please explain its reason.
Response #6
The design thickness mentioned here is the thickness when 100% of current efficiency is reached. Even though gold is a noble metal, but the current efficiency is still less than 100%. In addition, a stabilizer (sodium gluconate) is added into the electrolyte, which could also lower the current efficiency.
Comment #7
In equation 2, the comma should be deleted.
Response #7
A comma after the equation is a required format by MDPI Electrochem. In fact, there is also a comma at the end in equation 1.
Comment #8
Since micro-electrodeposition technology is employed in the paper, the following related papers should be cited in the introduction to electrochemical deposition (9-12).
Response #8
Thank you for the suggestions, we have added the two papers related to alloy plating into this manuscript.
Revision made
Reference #10 and #11
[10] Wang, C.; Bhuiyan, M.E.H.; Moreno, S.; Minary-Jolandan, M. Direct-write printing copper-nickel (Cu/Ni) alloy with controlled composition from a single electrolyte using co-electrodeposition. ACS Appl. Mater. Interfaces 2020, 12, 18683–18691.
[11] Bhuiyan, M.E.H.; Behroozfar, A.; Daryadel, S.; Moreno, S.; Morsali. S.; Minary-Jolandan, M. A hybrid process for printing pure and high conductivity nanocrystalline copper and nickel on flexible polymeric substrates. Sci. Rep. 2019, 9, 19032.

Reviewer 2 Report
This paper describes the effective Young’s Modulus of multilayered gold microstructures. The dependencies of structural variation and annealing temperature on the effective Young’s Modulus are indicated. These results will contribute to the field of micro-g sensor made of gold.
Reviewer to the authors
- Grammatical error: p. 2, l. 3 from the end: Fist “alao” should be eliminated.
- Descriptional error: Is “h” in Eqs. (1) and (2) “t” that stands for the thickness of cantilevers?
- Temperature measurement method in the experimental: In this paper, sample annealing temperatures are very important. You should state the way of temperature measurement of metal samples in a vacuum ambient more in detail. Because the actual temperature of metal is often largely different from the set-temperature depending on the way of heating/annealing, therefore, the heating/annealing methods such as using a gold image furnace, a lamp anneal or a block heater etc. and the temperature measurement methods such as using a thermo couple or a pyrometer, etc. should be stated in the experimental section. These will help the readers to understand the results.
Author Response
Comment
This paper describes the effective Young’s Modulus of multilayered gold microstructures. The dependencies of structural variation and annealing temperature on the effective Young’s Modulus are indicated. These results will contribute to the field of micro-g sensor made of gold.
Response
Thank you for the positive comments. We have revised the manuscript based on the comments.
Comment #1
Grammatical error: p. 2, l. 3 from the end: Fist “alao” should be eliminated.
Response #1
Thank you for pointing out the mistake.
Revision made
Line 82
………., which could affect the structure stability of micro-cantilevers.
Comment #2
Descriptional error: Is “h” in Eqs. (1) and (2) “t” that stands for the thickness of cantilevers?
Response #2
Thank you for pointing out the mistake.
Revision made
Line 155
……..where h and L and are the thickness and length of the fabricated……
Comment #3
Temperature measurement method in the experimental: In this paper, sample annealing temperatures are very important. You should state the way of temperature measurement of metal samples in a vacuum ambient more in detail. Because the actual temperature of metal is often largely different from the set-temperature depending on the way of heating/annealing, therefore, the heating/annealing methods such as using a gold image furnace, a lamp anneal or a block heater etc. and the temperature measurement methods such as using a thermo couple or a pyrometer, etc. should be stated in the experimental section. These will help the readers to understand the results.
Response #3
Thank you for the comment. The specimen annealed in this study is not pure metal. It is a piece of silicon substrate having patterns of electrodeposited gold micro-cantilevers on the surface. Therefore, it is mostly composed of silicon. We strongly agree that accuracy of the temperature applied to the micro-cantilever is important, therefore, the specimen was left at the desired temperature for 30 min to ensure proper annealing process. The temperature inside the vacuum chamber is constantly calibrated. On the other hand, we also agree that more info of the annealing process should be provided.
Revision made
Line 113
…….. annealed in a vacuum chamber (Heraeus Vacutherm Compact VT 6025, Thermo Scientific, Great Britain) at various temperatures ……
